Enhancing active ingredient biosynthesis in Chinese herbal medicine: biotechnological strategies and molecular mechanisms

Guo Ziyi 1
Yang Ning 2 15077823203@163.com
http://orcid.org/0000-0003-3695-2997 Xu Delin 1 2 xudelin2000@163.com
1 Department of Cell Biology, Zunyi Medical University , Zunyi, Guizhou , China
2 Department of Medical Instrumental Analysis, Zunyi Medical University , Zunyi, Guizhou , China
Zothanpuia
Electronic publication date: 2025 Feb 10
Publication date: 2025
Volume: 13
Electronic Location ID: e18914
Received 2024 Aug 27; Accepted 2025 Jan 7
Copyright: © 2025 Guo et al.
Copyright year: 2025
Copyright holder: Guo et al.
License: This is an open access article distributed under the terms of the Creative Commons Attribution License, which permits unrestricted use, distribution, reproduction and adaptation in any medium and for any purpose provided that it is properly attributed. For attribution, the original author(s), title, publication source (PeerJ) and either DOI or URL of the article must be cited.
License URL: https://creativecommons.org/licenses/by/4.0/

Keywords: Chinese herbal medicine (CHM), Active ingredients, Molecular regulation, Metabolic engineering, Biosynthesis

Funding: National Natural Science Foundation of China 32260089 Industry-University Collaborative Education Project of Ministry of Education 230901414190927 Future Outstanding Teachers Training Program of Zunyi Medical University XJ2023-JX-01-06 Postgraduate Teaching Reform Project of Zunyi Medical University ZYK105 Undergraduate Education and Teaching Reform Project of Zunyi Medical University XJJG2022-22, XJKCSZ2023-9, XJJG2024-09 Science and Technology Department Foundation of Guizhou Province QKHJC-ZK[2025]MS371, QKPTRC [2019]-027, QKHJC-ZK-2022-623 Class Advisor Studios at Zunyi Medical University 2024BZR-01 This research received financial support from the National Natural Science Foundation of China (32260089), the Industry-University Collaborative Education Project of Ministry of Education (230901414190927), the Future Outstanding Teachers Training Program of Zunyi Medical University (XJ2023-JX-01-06), the Postgraduate Teaching Reform Project of Zunyi Medical University (ZYK105) and the Undergraduate Education and Teaching Reform Project of Zunyi Medical University (XJJG2022-22; XJKCSZ2023-9, XJJG2024-09). The Science and Technology Department Foundation of Guizhou Province (Contracts No. QKHJC-ZK[2025]MS371, QKPTRC [2019]-027, QKHJC-ZK-2022-623) and the first batch of Class Advisor Studios at Zunyi Medical University (2024BZR-01) supported the APC of this article. The funders had no role in study design, data collection and analysis, decision to publish, or preparation of the manuscript.

==============================
Background

Chinese herbal medicine (CHM) is a fundamental component of traditional Chinese medical practice, offering a rich source of natural remedies with significant therapeutic potential. However, the scarcity of active ingredients and complex extraction procedures present substantial challenges to their widespread clinical application. This review aims to address this gap by exploring the potential of modern biotechnological advancements in enhancing the biosynthesis of these valuable compounds.

Methodology

The study takes a comprehensive approach, delving into the chemical composition of CHM’s active ingredients and elucidating their biosynthetic pathways and molecular regulatory mechanisms. Additionally, it surveys recent progress in extraction methodologies and evaluates engineering strategies aimed at synthetic production. This multifaceted analysis forms the foundation for examining the role of synthetic biology in augmenting CHM’s active ingredient synthesis.

Results

Our examination provides insights into the intricate biosynthetic pathways governing the formation of CHM’s active ingredients, as well as the complex molecular regulatory networks that underlie these processes. Furthermore, the review highlights advancements in extraction techniques, demonstrating their ability to streamline and enhance the isolation of these compounds. Engineering approaches for synthetic production, including metabolic engineering and synthetic biology tools, are assessed for their potential to overcome natural limitations and scale up production.

Conclusions

By integrating insights from biosynthesis, molecular regulation, extraction methodologies, and synthetic biology, this review establishes a robust theoretical framework for enhancing the production of CHM’s active ingredients. The proposed strategies and practical guidance aim to facilitate their broader utilization in modern medicine while promoting sustainability and accessibility within this invaluable medicinal heritage.

Introduction

Chinese herbal medicine (CHM) is a cornerstone of traditional Chinese medicine, with a rich history of clinical use and a vast repository of natural medicinal resources. CHM contains a wide array of active ingredients, including alkaloids, volatile oils, and flavonoidss (Shi et al., 2021), which have demonstrated significant therapeutic potentialt (Delfin, Watanabe & Tohge, 2019). Prominent examples include artemisinin and paclitaxel, which have made major contributions to drug development (Guo et al., 2023). However, the limited availability and complex extraction processes of these active ingredients hinder their widespread utilization in modern medicine (Hu et al., 2023).

Recent advances in biotechnology offer new opportunities for enhancing the synthesis of active ingredients in CHM. A comprehensive understanding of their structural characteristics, biosynthetic pathways, and molecular regulatory mechanisms is essential for efficient and artificial synthesis (Ma et al., 2020). Elucidating the chemical structure of an active ingredient enables the exploration of its biosynthetic pathway and molecular regulation, providing a theoretical foundation for biotechnological applications (Xu et al., 2021a). Research on biosynthetic pathways reveals the formation processes of active ingredients within plants, while the analysis of molecular regulatory mechanisms offers innovative strategies for precise regulation of biosynthesis (Li & Wu, 2018; Sun et al., 2023).

This review provides a comprehensive overview of the types, biosynthetic pathways, and molecular regulatory mechanisms of active components in CHM. It discusses engineering design strategies for the synthetic production of these active ingredients and explores the potential applications and limitations of relevant technologies in the conservation of medicinal plant resources and large-scale production of active ingredients (Li & Chen, 2024). Through this analysis and discussion, this paper aims to provide a theoretical foundation and practical guidance for the efficient synthesis of active ingredients in CHM, thereby facilitating their broader application and development in modern medicine.

Intended audience

This literature review is intended for researchers, academics, and professionals in plant biology, biotechnology, pharmacognosy, pharmacology, and traditional Chinese medicine (TCM). It provides a comprehensive overview of current advances in the biosynthesis pathways of secondary metabolites in medicinal plants, highlighting key enzymes, regulatory mechanisms, and genetic engineering strategies. Additionally, the review addresses regulatory challenges and ethical implications related to the biotechnological production of herbal medicines, making it valuable for regulatory bodies, policymakers, and interdisciplinary scholars. By consolidating recent studies and identifying research gaps, this manuscript serves as a comprehensive resource for stakeholders aiming to advance the integration of modern biotechnology with traditional medicinal practices and to inform ongoing and future research endeavors.

Survey methodology

In accordance with the Preferred Reporting Items for Systematic Reviews and Meta-Analysis (PRISMA) guidelines, a total of 214 articles were identified from PubMed (156), Google Scholar (31), and Scopus (27), and 37 articles were retrieved from other sources. After successive filtrations to eliminate irrelevant and duplicate studies (134), 117 highly relevant cutting-edge literatures were identified. Consequently, a systematic review and meta-analysis was conducted, incorporating 117 articles, which were analyzed and evaluated in depth to identify studies that promote the biosynthesis of the active ingredients of Chinese herbal medicines (Fig. 1).

Figure 1 PRISMA flow diagram.

Types and biosynthetic pathways of ingredients in CHM

CHM encompasses a diverse array of bioactive compounds (Cheng, 2021), primarily classified into alkaloids, saponins, flavonoids, polysaccharides, terpenoids, volatile oils, phenols, and organic acids (Lu & Jiang, 2013; Daley & Cordell, 2021; Wu et al., 2022; Islam et al., 2023). These compounds exhibit a spectrum of pharmacological activities, including antibacterial, anti-inflammatory, antioxidant, anti-tumor, immune-regulating, and cardiovascular protective effects (Chiș et al., 2023; Liu et al., 2017; Narayanankutty, Famurewa & Oprea, 2024; Zhang et al., 2023a; Zhou et al., 2016). The synergistic interactions among these compounds contribute to the holistic therapeutic effects of CHM (Zhao et al., 2023a).

The biosynthesis of secondary metabolites in plants involves complex metabolic pathways, including the acetate-malonic acid pathway, shikimic acid pathway, amino acid pathway, isoprene-like pathway, and polyamine biosynthesis pathway (Dawson et al., 2022; Javan et al., 2024; Rhee, Kim & Lee, 2007). These pathways utilize primary metabolites to synthesize a wide variety of complex secondary metabolites, underscoring the biochemical versatility that underpins the pharmacological diversity of CHM (Kishimoto et al., 2016; Zhang et al., 2023b).

Biosynthetic pathway of alkaloids

Alkaloids represent a significant class of active ingredients in CHM, renowned for their diverse pharmacological properties, including analgesic, antimicrobial, and anticancer activities. The biosynthesis of alkaloids is a highly studied area, with several key pathways elucidated and even reconstructed in heterologous hosts to enhance production scalability and sustainability (Zhou et al., 2023). For instance, tropane alkaloids such as cocaine; benzylisoquinoline alkaloids like berberine and sanguinarine; monoterpene indole alkaloids including brucine, strychnine, and iboga; and steroidal alkaloids such as alpha-tomatine exemplify the structural and functional diversity within this group (Jain, Tripathi & Tripathi, 2023; Lu et al., 2024; Wei et al., 2024). Despite significant advancements, the biosynthetic pathways of certain alkaloids, including diterpenoid alkaloids like aconitine, pyrrole alkaloids, and aristolochic acid—a benzylisoquinoline alkaloid—remain to be fully characterized.

Recent case studies have demonstrated the potential of biotechnological approaches in enhancing alkaloid production through meticulous gene identification, pathway assembly, and optimization. For example, the successful reconstruction of scopolamine, leonurine, and morphine biosynthetic pathways in heterologous hosts highlights the feasibility of using genetic engineering to achieve higher yields and sustainable production of these valuable compounds (Wicks, Hudlicky & Rinner, 2021) (Fig. 2). These advancements not only facilitate large-scale production but also pave the way for the discovery and utilization of novel alkaloids with therapeutic potential.

Figure 2 Biosynthetic pathway of three alkaloids.

Biosynthetic pathways of flavonoids

Flavonoids are synthesized through the phenylpropanoid pathway, which begins with the amino acid phenylalanine (Fig. 3). This precursor undergoes a series of enzymatic transformations, starting with the conversion of phenylalanine to cinnamic acid catalyzed by phenylalanine ammonia-lyase (PAL). Subsequently, cinnamic acid is hydroxylated to coumaric acid by cinnamate 4-hydroxylase (C4H), which is then transformed into chalcone by chalcone synthase (CHS) (Chen et al., 2019). Chalcone isomerase (CHI) catalyzes the isomerization of chalcone to naringenin, a pivotal intermediate that serves as a precursor for various flavonoid subclasses, including flavones, flavonols, flavanols, flavanones, and anthocyanins. This biosynthetic route not only underscores the structural diversity of flavonoids but also their significance in contributing to the pharmacological efficacy of CHM through their antioxidant and anti-inflammatory properties.

Figure 3 Biosynthetic pathways of flavonoids.

Biosynthetic pathways of terpenoids

Terpenoids are another major class of bioactive compounds in CHM, synthesized via two primary pathways: the mevalonate (MVA) pathway and the methylerythritol phosphate (MEP) pathway (Xu et al., 2021b) (Fig. 4). Both pathways generate the essential five-carbon (C5) building blocks isopentenyl pyrophosphate (IPP) and dimethylallyl pyrophosphate (DMAPP) (Gao et al., 2021; Upadhyay et al., 2018). These intermediates undergo condensation reactions to form larger prenyl diphosphates such as farnesyl pyrophosphate (FPP), geranyl diphosphate (GPP), and geranylgeranyl diphosphate (GGPP). These compounds are then subjected to cyclization and various enzymatic modifications, leading to the vast structural diversity of terpenoids (Chang, Li & Huang, 2018; Kuang et al., 2016; Wang et al., 2017). The intricate biosynthetic pathways of terpenoids not only contribute to the rich pharmacological profile of CHM but also offer numerous targets for metabolic engineering aimed at enhancing the production of specific terpenoid compounds with therapeutic benefits.

Figure 4 Biosynthetic pathways of terpenoids.

Molecular regulatory mechanism for the synthesis of pharmaceutical ingredients

The synthesis of pharmaceutical active ingredients is regulated by a complex interplay of molecular mechanisms, including signal transduction, enzyme catalysis, and gene expression regulation (Perez-Matas et al., 2024; Zheng et al., 2023).

Enzymes involved in biosynthesis of secondary metabolites

The biosynthesis of alkaloids, flavonoids, and terpenoids involves multiple enzyme-catalyzed steps. The activity, expression, and localization of these enzymes are tightly regulated (Zhou et al., 2021), influencing the production of secondary metabolites.

Key enzymes involved in alkaloid biosynthesis include scopolamine hydroxylase, putrescine N-methyltransferase, and N-methyl-putrescine oxidase (Qu, Safonova & De Luca, 2019). The discovery of these enzymes has provided insights into the regulation of alkaloid production (Jiang et al., 2024a).

Key enzymes in the flavonoid synthesis pathway include PAL, C4H, 4CL, and CHI (Li et al., 2018; Unuofin & Lebelo, 2020). CHS is a rate-limiting enzyme that catalyzes the production of naringenin chalcone, the precursor for downstream flavonoid synthesis (Zhong et al., 2022). CHI accelerates the isomerization of naringenin chalcone to naringenin, increasing the efficiency of the reaction (Lin, Chen & Dai, 2022).

Terpene synthases (TPS) catalyze the formation of diverse scaffolds from direct precursors (Li et al., 2023a). These scaffolds are then modified by dehydrogenases, reductases, glycosyltransferases, acyltransferases, and methyltransferases, resulting in the production of a wide variety of terpenoids with varying structures and biological activities (Jiang et al., 2024b; Li et al., 2023b; Yang & Drugs, 2017; Yang et al., 2023a).

Genes involved in the biosynthesis of pharmaceutical active ingredients

Gene expression plays a critical role in regulating the synthesis of bioactive drug ingredients. By controlling the expression of specific genes, intracellular metabolic pathways can be manipulated (Zhu, Yuan & Liu, 2024), thereby influencing the production of target molecules.

The genes involved in biosynthesis often belong to plant gene families, such as the cytochrome P450 (P450) family (Lv, Liu & He, 2017). Identifying specific genes within these families can be challenging due to gene expansion and contraction during evolution (Yang, Mao & Li, 2005). However, recent studies have revealed that genes involved in secondary metabolite biosynthesis tend to cluster together in the plant genome, forming metabolic gene clusters (Table 1) (Chen et al., 2023; Liao et al., 2022; Zuo et al., 2023). This gene clustering allows researchers to predict the biosynthesis pathways of secondary metabolites directly from genomic information. For example, the poppy genome contains a gene cluster for noscapine biosynthesis, which includes genes encoding O-methyltransferase and P450s (Miao et al., 2023; Rashid et al., 2023; Zhang et al., 2024a).

Table 1 Plant gene clusters and related genomes reported so far.

Time/year	Plant gene clusters	Related genomes	
1997	DIMBOA gene cluster		
2000		The genome of Arabidopsis thaliana	
2002		The genome of Oryza sativa	
2004	Momilactones and avenacins gene cluster		
2008	Thalianol gene cluster	The genome of Lotus japonicus	
2009	Phytocassanes and oryzalides gene cluster	The genome of Zea mays, Cucumis sativus and Sorghum bicolor	
2010		The genome of Ricinus communis	
2011	Linamarin, marneral, dhurrin and s lignin gene cluster	The genome of Selaginella moellendorffii	
2012	Noscapine gene cluster	The genome of Solanum lycopersicum	
The genome of Manihot esculenta	
The genome of Citrullus lanatus	
The genome of Hordeum vulgare	
2013	20-Hydroxybetulinic acid, arabidiol, α-tomatine, α-chaconine and α-solanine gene cluster		
2014	Casbene and cucurbitacins gene cluster	The genome of Brassica napus	
2015	Tirucalla-7,24-dien-β-ol and baruol gene cluster		
2016	β-Diketones and FPT gene cluster		
2018		The genome of Papaver somniferum	
2020	Acylsugar, falcarindiol, 5,10-diketo-casbene and Zx gene cluster	The genome of Solanum tuberosum	
2021	Hydroxycinnamoyl-tyramine and hydroxycinnamoyl-putrescine gene cluster	The genome of Ginkgo biloba	
2022		The genome of Alsophila spinulosa	
2023		The genome of Lonicera macranthoides	
2024	UFGT3 gene cluster	The genome of Fagopyrum esculentum Moench	

In addition to plant genes, it is increasingly recognized that many pharmaceutically relevant compounds traditionally isolated from plant materials are actually synthesized by symbiotic or endophytic fungi residing within the plant tissues (Schafhauser et al., 2019). These endophytic fungi possess their own biosynthetic pathways, enabling the production of complex bioactive metabolites such as paclitaxel, vincristine, and camptothecin (Birat et al., 2022; Subban & Kempken, 2023; Doan et al., 2024). This symbiotic relationship enhances the biosynthetic capabilities of both host plants and fungi, providing a sustainable and scalable source for valuable pharmaceutical compounds. Recent advancements in genomics and metabolomics have facilitated the identification of novel bioactive compounds produced by these microorganisms, expanding the repertoire of pharmacologically important substances and offering new avenues for biotechnological applications (Venugopalan & Srivastava, 2015; Hridoy et al., 2022; Tiwari & Bae, 2022).

Furthermore, transcription factors play a key role in regulating the biosynthesis of active medicinal ingredients. They modulate gene transcription rates and can be activated by various stimuli (Khoso et al., 2024; Zhao et al., 2023b). For instance, in the biosynthesis of artemisinin, a crucial antimalarial compound, transcriptional regulatory networks reveal intricate interactions that govern its production. Similarly, in Taxus chinensis, the MYB transcription factor TcMYB29a activates the expression of taxadiene-5α-hydroxylase (T5OH), enhancing taxol production (Gao, 2015; Liang et al., 2020). Salicylic acid (SA) can also boost paclitaxel yield by regulating specific transcription factors. In Salvia miltiorrhiza, the WRKY transcription factor SmWRKY2 activates SmCPS1 expression and promotes tanshinone production (Goyal et al., 2023; Javed & Gao, 2023; Kundu & Vadassery, 2021; Mahiwal, Pahuja & Pandey, 2024; Wani et al., 2021). Therefore, understanding the role of transcription factors in biosynthesis provides opportunities to manipulate gene expression and improve the production of pharmaceutical active ingredients (LaFountain & Yuan, 2021; Ren et al., 2023; Wang, Wu & Wei, 2024).

Effects of non-coding RNA on synthesis of pharmaceutical active ingredients

Additionally, non-coding RNA (ncRNA) plays a role in regulating the synthesis of pharmaceutical active ingredients (Zhan & Meyers, 2023). It affects the post-transcriptional regulatory system of plants, influencing the secretion and accumulation of secondary metabolites. Specifically, microRNAs (miRNAs) and small interfering RNAs (siRNAs) are the most common ncRNAs in plants (Li et al., 2024a). MiRNAs and siRNAs can identify target genes and degrade their messenger RNA, modulating gene expression. They are involved in the regulation of plant secondary metabolite biosynthesis. For example, miRNAs can enhance flavonoid biosynthesis by regulating MYB factor expression in plants (Hu et al., 2021).

In summary, the molecular regulatory mechanisms involved in the synthesis of pharmaceutical active ingredients are complex, encompassing enzyme catalysis, gene expression, regulation by transcription factors, and signal transduction. An in-depth analysis of the synthesis mechanisms of several key components within this system not only uncovers the intricacies of natural regulation but also paves the way for innovative approaches to optimize drug development strategies and enhance agricultural production efficiency, demonstrating unprecedented scientific value and application potential (Fig. 5).

Figure 5 The molecular regulatory mechanism of the synthesis of several crucial pharmaceutical active ingredients.

Engineering design of biosynthetic pharmaceutical ingredients

The artificial biosynthesis of pharmaceutical active ingredients is a comprehensive, interdisciplinary field encompassing biochemistry, molecular biology, chemical engineering, and other related disciplines (Ji et al., 2024; Zhang et al., 2014; Zhou et al., 2022). By selecting and expressing specific enzymes, providing precursor substances, regulating metabolic pathways, and optimizing reaction conditions, continuous improvement in synthesis strategies and key technical aspects (Chen et al., 2020; Deng et al., 2020; Huang et al., 2021) are expected to achieve efficient and customized production of target pharmaceutical active ingredients (Ding et al., 2023). This approach provides theoretical and technical support for transitioning from traditional reliance on natural resources to efficient, environmentally friendly, and sustainable industrial production modes for Chinese medicine active ingredients.

Main points of engineering design

Metabolic engineering and synthetic biology offer new approaches for producing pharmaceutical active ingredients. However, the complex biosynthetic pathways and low efficiency of heterologous synthesis remain challenges (Zhong et al., 2020). To address these challenges, it is essential to: (1) Mine and analyze biosynthetic genes for active components of traditional Chinese medicine. (2) Elucidate their biosynthetic pathways. (3) Optimize heterologous synthesis.

Novel chassis cells, such as Lipomyces starkeyi (Jin et al., 2023) and Methylobacterium extorquens (Yang et al., 2023b), combined with gene editing tools, can create efficient production systems for complex active ingredients. For example, researchers have developed a Pichia strain with high tyrosine productivity to synthesize tyrosine-derived compounds (Liu et al., 2023; Lu, 2021; Xia et al., 2023; Zhou et al., 2024).

Advancements in gene editing technologies, such as CRISPR-Cas, have enabled precise genetic modifications of chassis cells (Li & Xia, 2020). This technology allows for targeted gene insertion and deletion, regulation of gene expression, optimization of metabolic pathways.

Heterologous synthesis of pharmaceutical active ingredients

Heterologous synthesis entails the production of pharmaceutical active ingredients in host organisms that do not naturally synthesize these compounds. This biotechnological strategy provides precise control over the production process (Guo et al., 2022), mitigates environmental factors influencing natural biosynthesis (Du et al., 2024), and facilitates the generation of rare or geographically restricted compounds (Liu et al., 2019a; Yuan et al., 2024).

Advancements in metabolic engineering, gene editing, and synthetic biology have transformed the landscape of active pharmaceutical ingredient production, particularly for those derived from traditional Chinese medicine (TCM). By engineering hosts such as Escherichia coli and Saccharomyces cerevisiae, researchers have successfully replicated and optimized natural biosynthetic pathways to achieve higher yields and enhanced efficiency (He et al., 2024; Li et al., 2024b). A prominent example is the engineered yeast’s capacity to produce artemisinin, which significantly lowers costs while increasing availability compared to conventional plant extraction methods (Paddon et al., 2013).

From an economic perspective, heterologous synthesis diminishes reliance on variable agricultural outputs and alleviates risks associated with natural resource depletion. The scalability inherent in microbial fermentation guarantees a consistent and reliable supply of high-value pharmaceuticals, thereby addressing global healthcare demands (Zhang et al., 2024b). Clinically, manufacturing pharmaceuticals within controlled environments enhances compound quality and purity by minimizing contaminants, thus ensuring greater patient safety (Ye et al., 2024).

Successful applications include the genetic modification of E. coli for resveratrol production aimed at cardiovascular health benefits, as well as berberine biosynthesis in S. cerevisiae—an alkaloid recognized for its antimicrobial and anti-inflammatory properties (Han & Li, 2023; Liu et al., 2024). These case studies illustrate both the versatility and practical significance of heterologous synthesis in developing innovative therapies rooted in traditional medicinal practices.

In conclusion, heterologous synthesis plays a crucial role in contemporary biotechnology for producing pharmaceutical active ingredients. Its capability to enhance production efficiency, ensure sustainability, and support the development of high-quality medicinal compounds renders it indispensable for advancing both traditional healing practices and modern healthcare solutions.

Joint strategies of synthetic biology and metabolic engineering

The convergence of synthetic biology and metabolic engineering has revolutionized modern medicine by addressing challenges inherent in traditional drug production. These technologies enable cost-effective and resource-efficient production while promoting innovation in manufacturing processes to achieve environmentally sustainable, economical, and efficient outcomes, which promise to be applied in the production of Chinese herbal active ingredients (Qiu et al., 2024).

For plant alkaloids, the biosynthesis and identification of regulatory elements have advanced the molecular breeding of medicinal plants, facilitating the heterologous production of medicinal alkaloids (Zhao et al., 2020). Genome analysis provides insights into the evolutionary processes governing alkaloid biosynthesis genes, informing the selection of appropriate pathways for synthesizing targeted alkaloids within plants. This considers the complexity and distribution of metabolic networks and the challenges of establishing genetic transformation systems.

For flavonoids, recent studies have established the therapeutic potential of flavonoids derived from fruits, vegetables, and medicinal plants (Ding et al., 2023). However, traditional production methods, such as plant extraction and chemical synthesis, face scalability limitations for mass industrial production. Advanced genetic engineering tools enable the modification of biosynthetic pathways in plants, microorganisms, and industrially significant entities, expanding opportunities for large-scale flavonoid production (Huang et al., 2023). Co-culture engineering approaches leverage the capabilities of multiple engineered microbial strains to reconstruct targeted biosynthetic pathways, overcoming the constraints of traditional monocultures.

For terpenoids, metabolic engineering has been extensively explored in cyanobacteria, bacteria, and yeast. Englund et al. (2018) demonstrated the potential of cyanobacteria as a robust chassis for synthetic biology, achieving high isoprene yields through overexpression of genes in the MEP pathway and evaluating the influence of metabolic pathways such as the Calvin cycle and glycolysis on terpenoid synthesis (Englund et al., 2018; Sahu et al., 2023).

Plant metabolic engineering for detoxification of plant toxins involves transferring regulatory mechanisms governing plant secondary metabolite synthesis into heterologous systems for precise control. Synthetic biology strategies precisely modulate the localization of secondary metabolites to enhance plant growth and stress responses. Researchers have successfully rewired plant hormone pathways to manipulate plant development and structure (Chao et al., 2019). Additionally, plant-microbe interactions mediated through metabolites present opportunities for engineering efforts aimed at improving plant growth and resilience.

Discussion and prospect

Despite significant advancements in understanding the biosynthetic pathways and regulatory mechanisms of active ingredients in CHM, considerable gaps persist in CHM active ingredient research. Current progress underscores the necessity for intensified exploration of biosynthetic pathways, refinement of metabolic control, accelerated research through novel technologies, and the development of innovative production strategies. This forward-looking roadmap holds substantial promise for unlocking CHM’s full potential in health promotion and sustainable development.

In-depth understanding of biogenic synthesis pathways and regulatory mechanisms

A comprehensive understanding of the biosynthetic pathways and regulatory mechanisms of active ingredients in CHM is crucial for enhancing their yield and quality. Systems biology approaches, including genomics, transcriptomics, and metabolomics, have provided detailed maps of these pathways, identifying key enzymes and regulatory factors (Ma et al., 2015). For instance, genomics has facilitated the identification of gene clusters responsible for secondary metabolite production, while transcriptomics has elucidated gene expression patterns under various environmental conditions. Metabolomics complements these studies by profiling the metabolites involved in these pathways, offering insights into the dynamic changes during synthesis.

However, these methodologies are not without limitations. Genomic studies often face challenges related to the complexity of plant genomes and the presence of gene families with redundant functions. Transcriptomic analyses can be confounded by the spatial and temporal variability of gene expression, making it difficult to pinpoint specific regulatory mechanisms. Metabolomics, while powerful, may miss low-abundance metabolites or those that are rapidly metabolized. Additionally, epigenetic regulation studies, which explore DNA methylation, histone modification, and non-coding RNA impacts, provide valuable information but are still in the nascent stages concerning CHM.

Research on microbial-plant interactions, particularly the role of rhizosphere microbiota and endophytes in biosynthetic synthesis, offers another layer of complexity. While these interactions can enhance the production of certain metabolites, they also introduce variability that complicates the reproducibility of CHM preparations. Understanding the spatio-temporal regulation mechanisms of intercellular metabolic flow remains a significant challenge, as environmental factors and developmental stages intricately influence biosynthetic processes.

Precise regulation of secondary metabolic pathways

With a foundational understanding of biosynthetic pathways and regulatory mechanisms, precise regulation of secondary metabolic pathways can be achieved to enhance the yield and quality of active ingredients. Metabolic engineering techniques, such as the overexpression of key enzymes or regulatory factors and the knockout of inhibitory elements (Liu et al., 2019b), have shown promise in boosting biosynthetic production. Synthetic biology approaches further enable the design of artificial biosynthetic pathways by introducing heterologous genes or optimizing enzyme activity, thereby tailoring the production of specific metabolites. Monocellular technology provides a more refined approach by allowing the investigation of cell or tissue-specific secondary metabolic pathways and product accumulation mechanisms. This precision facilitates targeted pathway regulation, optimizing CHM production to meet increasing market demands.

However, these strategies also present challenges. Metabolic engineering often requires extensive trial and error to identify optimal gene combinations, and synthetic biology approaches may encounter issues related to the stability and scalability of engineered pathways. Additionally, monocellular technologies need to balance precision with the inherent biological variability of living systems, which can affect the consistency of metabolite production.

Use of new technologies to accelerate the research process

Novel technologies offer powerful tools for studying the biosynthetic pathways and regulatory mechanisms of active ingredients in CHM. High-throughput sequencing technologies, such as RNA-Seq and advanced metabolomics platforms, enable rapid and comprehensive analysis of gene expression and metabolite profiles. Microscopic techniques, including confocal and electron microscopy, allow for the visualization of organelle and molecular interactions within biosynthetic pathways (Yuan et al., 2023). Furthermore, artificial intelligence (AI) and machine learning (ML) techniques facilitate the analysis of large datasets, identifying patterns and predicting key factors in biosynthesis pathways.

While these technologies significantly accelerate research, they also introduce new challenges. The sheer volume of data generated by high-throughput methods requires robust computational infrastructure and sophisticated data analysis algorithms to extract meaningful insights. AI and ML models, although powerful, depend heavily on the quality and quantity of input data and may produce biased or inaccurate predictions if not properly validated. Additionally, advanced microscopic techniques necessitate specialized expertise and equipment, potentially limiting their accessibility for some research groups.

Explore new ways of production

The biosynthesis of active ingredients in CHM has catalyzed new avenues for drug development. A deeper understanding of biosynthetic pathways and regulatory mechanisms enables the design and synthesis of new analogues with improved biological activity, reduced toxicity, and enhanced stability. Synthetic biology facilitates the construction of complex CHM active ingredients that are otherwise inaccessible through traditional extraction methods. Moreover, regulating biosynthetic pathways can identify novel therapeutic targets, contributing to innovative therapies for diseases such as metabolic disorders and cancer. For example, targeting glutaminase (GLS), a key enzyme in glutamine metabolism, has been shown to inhibit tumor growth and enhance immunotherapy effectiveness in certain cancers (Li, Ng & Ng, 2023).

However, these biotechnological advancements are not without inherent limitations and risks. The introduction of genetically modified organisms (GMOs) for the production of CHM compounds raises ecological and ethical concerns, including potential unintended impacts on ecosystems and biodiversity. Additionally, the synthesis of novel analogues may result in unforeseen pharmacological effects, necessitating extensive safety evaluations. Regulatory frameworks must evolve to address these challenges, ensuring that biotechnological innovations in CHM are both safe and sustainable.

Critical evaluation of methodologies and future directions

While the integration of synthetic biology and metabolic engineering holds immense potential for the synthetic synthesis of active CHM ingredients, it is essential to critically assess the methodologies employed. Traditional extraction methods, though time-tested, often suffer from low yields and limited scalability. In contrast, biotechnological approaches promise higher efficiency and sustainability but require significant investment in research and development. The scalability of engineered pathways from laboratory conditions to industrial production remains a hurdle, as does the need for cost-effective and robust bioprocessing technologies.

Future research should prioritize uncovering additional biosynthetic pathways of Chinese herbal active ingredients, developing more efficient detection techniques, and enhancing computational methods for pathway prediction and optimization. Furthermore, optimizing synthetic strategies to improve the stability and yield of engineered pathways is crucial. Addressing these technical challenges will not only support sustainable agriculture and pharmaceutical research but also contribute significantly to human well-being by providing reliable and effective CHM-based therapies.

In conclusion, while the advancements in biotechnological applications for CHM are promising, a balanced approach that critically evaluates methodologies, acknowledges inherent limitations, and addresses potential risks is imperative. By comprehensively understanding and optimizing biosynthetic pathways and regulatory mechanisms, and by leveraging new technologies judiciously, the full potential of CHM can be harnessed to benefit global health sustainably.

Conclusions

The large-scale biotechnological production of Chinese herbal medicines (CHM) presents significant regulatory and ethical challenges that must be addressed to ensure sustainable and responsible implementation. Variations in regulatory frameworks across regions necessitate harmonization to facilitate international collaboration and market access. Ensuring consistent quality and efficacy requires stringent quality control measures and standardized protocols. Comprehensive safety and efficacy assessments, including clinical trials, are essential to validate biotechnologically produced CHM.

Ethical issues related to intellectual property and bioprospecting must be managed to protect traditional knowledge and ensure fair benefit distribution, preventing biopiracy and promoting equitable access to genetic resources. Additionally, the environmental impact of large-scale production and the socioeconomic effects on communities involved in CHM cultivation must be carefully considered to maintain biodiversity and support local livelihoods. Ethical considerations in genetic engineering, such as maintaining ecosystem balance and addressing the long-term effects of modified traits, are also critical.

Addressing these challenges through interdisciplinary collaborations, the development of comprehensive guidelines, and promoting transparency and stakeholder engagement will support the responsible and ethical advancement of biotechnological innovations in CHM.

Supplemental Information

Supplemental Information 1 PRISMA 2020 checklist.

The authors deeply appreciate the anonymous reviewers for their incisive comments, which have significantly enriched the substance and presentation of this literature review. Furthermore, the utilization of pivotal tools such as EndNote for citation management; PubMed, Google Scholar, and Web of Science for comprehensive research access; and Grammarly Editor for expert writing guidance has been instrumental in crafting a meticulous and thorough review, thereby advancing the scholarly conversation in our field.

Additional Information and Declarations

Competing Interests

The authors declare that they have no competing interests.

Author Contributions

Ziyi Guo performed the experiments, analyzed the data, prepared figures and/or tables, and approved the final draft.

Ning Yang analyzed the data, prepared figures and/or tables, authored or reviewed drafts of the article, and approved the final draft.

Delin Xu conceived and designed the experiments, authored or reviewed drafts of the article, and approved the final draft.

Data Availability

The following information was supplied regarding data availability:

This is a literature review.

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
