# Peer review of "Enhancing active ingredient biosynthesis in Chinese herbal medicine: biotechnological strategies and molecular mechanisms"

_PeerJ, doi:10.7717/peerj.18914_

## Round 0.1 · original submission · Minor Revisions

The authors should include the suggestion of the reviewers

Reviewer 1 ·

Basic reporting

-The article is well-written with clear, professional language. The introduction effectively sets the stage for the study by highlighting the challenges of biosynthesis in Chinese Herbal Medicine (CHM) and the potential of biotechnological strategies.
- Further in some sections, minor grammatical refinements would improve fluency.
-The authors should also include recent papers that reflect cutting-edge biotechnological developments in CHM to support the article’s conclusions.
- The figures are relevant and clear. No inappropriate image manipulation was observed. They appropriately represent complex pathways and mechanisms.

Experimental design

- The article fits well within the scope of the journal, focusing on enhancing biosynthetic pathways of CHM through biotechnological methods, which is both relevant and timely. The subject is of cross-disciplinary interest, bridging traditional medicine, biotechnology, and molecular biology.
- A more detailed methodology on how these pathways are reconstructed in heterologous hosts should be included. For example, when discussing alkaloid biosynthesis (Section 1.1), further explanation of how these pathways can be practically applied in industrial-scale CHM production would be useful.
- The article follows a systematic review structure, adhering to PRISMA guidelines and providing a comprehensive analysis of over 100 sources. The methodological rigour is evident.
- More emphasis on comparing traditional extraction methods with these biotechnological advances could strengthen the study design. Highlight how the proposed strategies overcome specific limitations like the low yield or instability of certain CHM compounds.

Validity of the findings

- The conclusions are well-supported by the data provided, but more focus is needed on the practical implications of biotechnological interventions in CHM.
- The article could benefit from a more nuanced discussion on the potential economic or clinical implications of these biotechnological strategies. For instance, expanding on the section on synthetic biology (Section 3.2) to include more case studies of successful implementation would enhance the real-world relevance.
- The identification of gaps and future research directions is commendable, particularly in addressing the need for novel technologies and interdisciplinary collaboration. However, the article could better explore the regulatory challenges or ethical considerations related to large-scale biotechnological production of CHM.

·

Basic reporting

The article is generally well-written with proper English usage. Minor grammar improvements could enhance readability and ensure clarity.

The background is comprehensive, providing a broad context on Chinese Herbal Medicine (CHM) and relevant biotechnological advancements. The literature cited appears up-to-date and relevant, supporting the discussion on CHM biosynthesis and molecular regulation.

The structure aligns well with standard review formats. The manuscript flows well but sometimes repeats concepts, particularly within the sections on biosynthesis pathways.

Experimental design

The study is well-designed, fitting the journal's aims and scope, and follows PRISMA guidelines, which adds transparency.

Given the title of this review, there should be a stronger critical comparison of methodologies among the studies referenced, rather than a descriptive overview. Identifying specific strengths and limitations would enhance the rigour of the review.

Validity of the findings

The findings are thoughtfully synthesised, providing a comprehensive overview of current advancements in CHM biosynthesis. Although the manuscript outlines future directions, it primarily emphasises the benefits of biotechnological applications without sufficient attention to the inherent limitations and risks.

Additional comments

Table 1 seems incomplete? Please check.

Reviewer 3 ·

Basic reporting

It is a review of a broad interest about the current knowledge on active ingredient biosynthesis process in Chinese herbal medicine plants and biotechnological strategies and molecular mechanisms for production of relevant metabolites. Overall the paper is comprehensive and contains analysis of available references providing enough information to the relevant field. I would reccoment following optimizations:

Line 86 - suggest to add: "and polyamines" (https://doi.org/10.1111/j.1582-4934.2007.00077.x)
Line 129 - enzyme -> enzymes
Line 146 - please add/explain in the section information that many pharmaceuticaly relevant compounds are isolated from plant material, but are actually synthesized by the symbiotic/endophytic fungi (e. g. DOI: 10.1073/pnas.1910527116)
Line 302-303 - please add a reference with an example
Line 318 - a double space is located between "as" and "Pubmed"

English - minor proof-reading is recommended.

On all other criteria the article is in compliance with the journals criteria.

Experimental design

The article is in compliance with the journals criteria.

Validity of the findings

The article is in compliance with the journals criteria.

Additional comments

No additional comments.

---

## Round 0.2 · accepted · Accept

The authors have addressed all of the reviewers' comments, and the manuscript is ready for publication

Reviewer 1 ·

Basic reporting

The introduction needs strengthening by including the previously suggested references. Hence, I strongly recommend to follow my previous report for the relevant references, update the manuscript.

Experimental design

It's improved as per suggestions

Validity of the findings

It's fine

·

Basic reporting

The manuscript provides a clear and comprehensive overview of the biotechnological strategies employed to enhance active ingredient biosynthesis in Chinese herbal medicine. The contents well-structured, with sufficient context and references to support the study. Figures and tables are appropriate and add value to the content.

Experimental design

The authors have responded well to all raised issue and have implemented the necessary corrections.

Validity of the findings

The findings are credible and well-supported by existing literature.

Additional comments

The authors have responded well to all raised issue and have implemented the necessary revisions. The manuscript is suitable for publication in its current form.

Reviewer 3 ·

Basic reporting

Authors addressed all reviewers comments in the revised version of the paper and optimized the manuscript accordingly

Experimental design

Authors addressed all reviewers comments in the revised version of the paper and optimized the manuscript accordingly. Study design is sufficient and the content is within the aim and scope.

Validity of the findings

Authors addressed all reviewers comments in the revised version of the paper and optimized the manuscript accordingly. The findings are valid, conclusions are well stated.